# Aspect-Based Multimodal Mining: Unveiling Sentiments, Complaints, and Beyond in User-Generated Content

## ABSTRACT

Sentiment analysis and complaint identification are key tools in mining user preferences by measuring the polarity and breach of expectations. Recent works on complaint identification identify aspect categories and classify them into complaint or not-complaint classes. However, aspect category-based complaint identification provides high-level information about the features of products. In addition, it is also observed that the user sometimes does not complain about a specific aspect but expresses concern about specific aspects in a respectful way. Currently, uni-modal and multimodal studies do not differentiate between this thin line between complaint and concern. In this work, we propose the task of multimodal aspect term-based analysis beyond sentiments and complaints. It comprises of two sub-tasks, *viz* (i) classification of the given aspect term into one of the four classes, *viz.* praise, concern, complaint, and others, (ii) identification of the cause of praise, concern, and complaint classes. We propose a first benchmark explainable multimodal corpus annotated for aspect term-based complaints, praises, concerns, their corresponding causes, and sentiments. Further, we propose an effective technique for the joint learning of aspect term-based complaint/concern/praise identification and cause extraction tasks (primary tasks) where sentiment analysis is used as a secondary task to assist primary tasks and establish them as baselines for further research in this direction. [1]

## CCS CONCEPTS

• **Do Not Use This Code → Generate the Correct Terms for Your Paper**; *Generate the Correct Terms for Your Paper*; Generate the Correct Terms for Your Paper; Generate the Correct Terms for Your Paper.

## KEYWORDS

Do, Not, Us, This, Code, Put, the, Correct, Terms, for, Your, Paper

## 1 INTRODUCTION

Online user reviews have become a critical source of information across a broad spectrum of domains, including e-commerce, hospitality, healthcare, and entertainment [11, 16, 35]. Nowadays, users

---

[1] Sample dataset has been made available at: https://anonymous.4open.science/r/MAspectX-327E/README.md The whole dataset will be made publicly available for research after acceptance of the paper.

**Unpublished working draft. Not for distribution.**

are posting both visual and textual information in order to better express their opinions and experiences. Organizations can improve their products or services by mining rich-source textual and visual content. Additionally, it allows other customers to make more informed decisions about their purchases.

Sentiment analysis and complaint detection play an important role in accomplishing these objectives [12, 21, 33]. Sentiment analysis classifies the polarity of a review into one of the three classes, *viz.*, positive, negative, and neutral [17]. However, due to limited information captured by sentiment analysis systems, researchers proposed the task of complaint detection, which identifies breaches of expectation (complaints) from user-generated reviews. Complaint identification is a binary classification problem, categorizing review as a complaint or not a complaint [21].

There have been works on complaint identification at the document or the sentence levels [4, 8, 12]. In the prior works, authors have also identified the severity [8, 32] and cause [26, 27] of complaints at the sentence level. An attempt has been made towards analyzing complaints at a fine-grained level [25], using a multimodal corpus (text + image) for binary complaint identification of each aspect category described in the review. Although this corpus provides vital information at the aspect level, it neither provide specific and fine-grained information about specific features or attributes (aspect terms) nor their cause for particular aspect . For example, the customer may complain about the *neck size* but can be happy with the *arm size* in the same review. However, authors in [25] only provide information that the user is complaining about the *size*. It does not reveal whether the user complains about *neck* or *arm size* (aspect terms) and the cause for their dissatisfaction. This missing information can mislead and hinder the organizations in their decision-making processes, as there can be multiple aspect terms falling under one aspect category.

Apart from expressing satisfaction (praise) or dissatisfaction (complaints), we also observe that sometimes reviewers do not complain about a specific aspect but express a feeling of worry or interest about something, often with the intention of improving an issue. We call it a *concern.* Concern is usually expressed in a constructive and respectful way, focused on finding a solution or improvement or making suggestions to others. For example, *Be careful with what size you order; those who are looking for comfort can order for 1 size larger than usual.* Here, the user is suggesting to be careful with what size they order, hence expressing a concern for size. Distinguishing between concerns and complaints can help in managing customer relationships. Addressing concerns promptly and constructively can enhance customer satisfaction, while efficiently resolving complaints can help the dissatisfied customers.

However, to the best of our knowledge, current studies (uni- and multimodal) do not consider this differentiation between concerns and complaints and also do not provide complaint analysis and corresponding cause extraction at the aspect term level. To bridge this

gap, we propose a new task, namely, aspect term based multimodal analysis beyond sentiments and complaints. Our proposed task consists of two sub-tasks, *multimodal aspect term based analysis beyond sentiment and complaint (MAspectX)* and *finding the cause of class predicted in first sub-task (MAspectX-Cause)*. Given the review and image, *MAspectX* task aims to classify the given aspect term into one of the four classes, *viz.*, complaint, concern, praise, and others. Given the review and image, *MAspectX-Cause* task aims to extract the cause behind the class *MAspectX* for a given aspect. The cause or span of the *MAspectX* class can vary. It could be a complete sentence, phrase, or even a single word. Few examples are presented in Figure 1.

As there is no such corpus available, we first build a multimodal corpus (MAspectX) using large vision models to solve both the tasks [15, 31]. Further, To assist the *MAspectX* task, the dataset is also labeled for sentiment labels for every aspect term mentioned in the review. This task is known as aspect term sentiment analysis (MATSA) [19, 20]. Figure 1 shows a few example review texts and images with the corresponding MATSA, MAspectX, and MAspectX-Cause labels. In the examples, it is observed that text and images together provide a better understanding of the labels compared to only uni-modal.

The key contributions of our current work are as follows:

- We propose a new task multimodal aspect term based analysis beyond sentiment and complaint, *MAspectX*, to classify the aspect terms into one of four classes, *viz.* praise, complaint, concern, and others.
- We propose multimodal aspect term based cause extraction task, *MAspectX-Cause* to identify the rationale or causes behind the complain/concern/praise/other classes for every aspect term.
- We develop a new benchmark dataset for *MAspectX* and *MAspectX-Cause* tasks in a semi-supervised fashion using multi-step few-shot prompting.
- We build a multi-task system to jointly solve *MAspectX*, *MAspectX-Cause* and *MATSA* tasks in parallel.
- The proposed model outperforms the state-of-the-art methods and can serve as a benchmark for further research in this direction.

## 2 RELATED WORK

Previous works on complaint detection identify the complaints at the document or the sentence levels using feature-based machine learning models [4, 12] and transformers-based models [22]. Additionally, the multitask model has been proposed to aid in complaint analysis by incorporating polarity and affect information [28]. Efforts have also been directed towards identifying the severity [8, 32] and cause [26, 27] of complaints. However, all these attempts to identify severity and cause at the sentence level. In the existing literature, complaints have been categorized on the basis of several factors, including the responsible department, product hazards, degree of urgency, and risks [2, 13, 36]. Recently, attempts have been made towards analyzing complaints at a fine-grained level [25, 29], using a multimodal corpus (text + image). This corpus is annotated for complaint and non-complaint labels for each aspect category described in the review. However, this data does not

provide any specific or more fine-grained information about the specific features or attributes (aspect terms) and corresponding cause (explainability).

Aspect-based complaint analysis (*ABCA*) shares similarities with the well-established domain of aspect-based sentiment analysis (ABSA). However, it is crucial to recognize that despite their connections, *ABSA* and *ABCA* are distinct concepts with different objectives, scopes, and outputs. While, ABSA helps in understanding how customers feel about various aspects of a product or service [10, 34, 37], *ABCA*, on the other hand, focuses on identifying and understanding complaints made by customers.

Our work addresses the following limitations of current uni-modal and multimodal studies: (i) current studies lack the capability to conduct complaint analysis at the aspect term level; (ii) They fail to differentiate effectively between complaints and concerns; (iii). existing studies do not provide the cause for complaints at a fine-grained level.

To address these limitations, we propose new tasks, *MAspectX* and *MAspectX-Cause* to classify the aspect terms into one of the four classes and identify the cause behind them. In addition, we leverage the information from the sentiment task to enhance the performance of our proposed tasks.

## 3 RESOURCE CREATION

To create our corpus, we scraped reviews from Amazon's clothing domain, specifically targeting multimodal reviews in English[2]. We follow a multi-step approach to annotate our corpus to reduce the manual annotation efforts. Verifying existing labels is generally quicker and more cost-effective than starting the labeling process from scratch. It reduces the need for annotators to label entire datasets and instead focuses on confirming or correcting existing annotations.

The recent success of pre-trained large-scale models motivates us to utilize them for data annotations [15]. Multi-step few-shot prompting is used to assign weak labels to the dataset, where we use the Large Language and Vision Assistant Pretrained Transformer (LLaVA) model for prompting[3]. It uses a relatively small set of in-context examples (about 10 random samples).

### 3.1 Multi-step Few Shot Prompting

We have three tasks, namely *MATSA*, *MAspectX*, and *MAspectX-Cause*. As a first step, we first extract all possible aspect terms present in the review sentence. The output of this step is passed as input to the *MATSA*, *MAspectX*, and *MAspectX-Cause* tasks. Hence, three distinct prompts have been designed, each tailored for a specific step of the annotation process.

**Step 1: Extraction of Aspect Terms:** We construct the following prompt template to ask LLaVA about the possible aspect terms mentioned in the review sentence.

*F1 [Given the review R], which specific aspect terms are possible mentioned?*

Here, *F1* are the few shot examples provided for the ATE task (consisting of text and image). Let, *A* be the answer given by LLaVA,

---

[2]https://www.amazon.in
[3]https://llava-vl.github.io/

| Review | Images | Aspect terms | Labels | Reason for each MaspectX class |
|---|---|---|---|---|
| There was a nearly inch hole in the kurti. Quality of palazo is good. | | kurti, Quality of palazo | **MAspectX**: complaint, praise

**MATSA**: negative, positive | [There was a nearly inch hole in the kurti], [Quality of palazo is good] |
| The material is thick so it's not transparent at all. I think it's worth the price. | | meterial, price | **MAspectX**: praise, praise

**MATSA**: positive, positive | [The material is thick so it's not transparent at all], [worth the price] |
| Price tags missing. | | Price tags | **MAspectX**: complaint

MATSA: negative | [Price tags missing] |
| Disappointed to see the inner fabric | | inner fabric | **MAspectX**: complaint

**MATSA**: negative | [Disappointed to see the inner fabric ] |
| Style is too good , but it should be half sleeves | | Style, sleeves | **MAspectX**: praise, concern

**MATSA**: positive, neutral | [Style is too good], [it should be half sleeves] |

Figure 1: Examples for *MAspectX, MATSA, MAspectX-Cause* tasks. Terms and corresponding labels, all are separated by a comma.

containing all possible aspect terms (*asp*) mentioned in the review *R*. If no aspect term is mentioned in *R*, the value of *A* is *no mention*.

**Step 2: MATSA:.** For a given aspect term $asp_j$ (obtained from step 1), we construct the following prompt template to ask LLaVA about the possible sentiment class of aspect term among the possible three classes, *viz*. positive, negative, and neutral.

*F2 [Given the review S], classify the aspect $asp_j$ into one of the three classes, positive, negative, and neutral.*

Here, *F2* are the few shot examples provided for the ATSA task (text and image).

**Step 3: MAspectX and MAspectX-Cause:** For a given aspect term $asp_j$ (obtained from step 1), we construct the following prompt template to obtain the possible class of $asp_j$ among the 4 classes, praise, complaint, concern, and others and cause behind this classification.

*F3 [Given the review R], classify the aspect $asp_j$ into one of the four classes: complaint, concern, praise, others, and extract the phrase responsible for the predicted class from the review.*

Here, *F3* are the few shot examples for MAspectX and MAspectX-Cause tasks (text and image). Suppose there are *k* number of aspect terms in the sentence, then steps 2 and 3 are repeated for each of the aspect terms ($asp_j$).

## 3.2 Manual Correction

To maintain the quality of the labels, we manually analyzed the labels obtained using a multi-step few-shot prompting method. We also observed a few errors in the labels. The majority of the errors were in the identification of aspect terms. For example, sometimes, LLaVA remembers the aspect terms provided in the few-shot context. Hence, it assigns the same aspect term to the current sentence,

regardless of whether the review is talking about that attribute or not. For example,

**Input:** Neck size is perfect, shoulder size is big.

**Output:** Neck size, shoulders, price.

Here, price is not part of the current sentence, but LLaVA has assigned it because of the few-shot context. Therefore, to ensure the annotation quality of the dataset, we engaged three linguists with sufficient subject knowledge and experience in the construction of supervised corpora. Two have doctoral degrees, and one has completed his master's. Linguists are asked to manually verify the annotations of every step. After the first step of annotation, linguists have verified the aspect term labels, so missing aspect terms can be added to the labels, or additional terms can be removed. This step ensures the quality of labels obtained at subsequent steps.

After verification of first step labels, second and third step were performed to reduce the error rate. Guidelines along with some examples were explained to the linguists before starting the verification process. We follow the guidelines used in SemEval aspect level sentiment analysis task for verification of the aspect terms and our MATSA task [19]. We use the complaint definition from an earlier linguistic study: "A complaint presents a state of affairs that breaches the writer's favorable expectation" [3]. In addition to satisfaction (praise), we introduced a new class *concern*. MAspectX-Cause is a portion of text that express why the customer feels satisfaction, dissatisfaction, expressing a concern or others. We attained an overall Fleiss' kappa [30] score of 0.75 and 0.79 among the three linguists for MATSA and MAspectX tasks, respectively. For *MAspectX-Cause* task, we measure the quality of annotations using macro-F1 following earlier research [23, 27] and it comes out to be 0.78. These agreement scores indicate that the annotations can considered as reliable. The use of weak labeling prior to actual

annotation reduced the annotation time and also helped in achieving the good quality labels. A few samples along with their labels are shown in Figure 1.

**Data Distribution:** The *MAspectX* dataset contains a total of 4,966 aspect terms with 3303 positive, 981 negative, and 682 neutral aspect instances. Detailed class-wise distribution of aspects for *MATSA* and *MAspectX* tasks are shown in Table 1. There is a strong dominance of the positive/praise classes among the other classes. Instances for complaint/concern classes are comparatively less than other classes, which is in line with other works [24].

**Table 1: Data Distribution of MAspectX Corpus.**

| Task | Complaints | Concern | Praise | Others | Total |
|------|-----------|---------|--------|--------|-------|
| MAspectX | 732 | 306 | 3303 | 625 | 4966 |
| | **Positive** | **Negative** | **Neutral** | | **Total** |
| MATSA | 3303 | 981 | 682 | - | 4966 |

## 4 METHODOLOGY

We develop a multimodal multi-task model for MATSA, MAspectX, and MAspectX-Cause tasks for a given aspect term (*MMAspectX*). The system employs a shared fusion mechanism to infuse the features from the multimodal inputs. Detailed architecture of our proposed *MMAspectX* is depicted in Figure 2. The review text and aspect term are given as input to the text encoder (BERT), and the corresponding image is fed into the visual encoder (ResNet). Further, AutoEncoder component enhances the model's ability to capture subtle nuances and opinions expressed in the input text. Finally, the shared fusion component integrates information from both textual and visual modalities to provide a comprehensive understanding of the input data.

### 4.1 Text Encoder

To encode the textual modality, our textual encoder uses BERT (Bidirectional Encoder Representation from Transformers) [5]. BERT encoder has been pre-trained on two unsupervised tasks using a large Wikipedia corpus to capture the contextual relationship between words and sentences. Input to BERT encoder is *review [SEP] $asp_j$*, where $asp_j$ is the $j^{th}$ aspect of the multimodal review.

### 4.2 Image Encoder

We employ ResNet[4] to capture rich visual cues from the images. To simplify the problem and maximize the utility of the embedding space, we partition the embedding dimensions and image data into distinct groups. Each learner is responsible for creating a unique distance metric using a subspace of the original embedding space and a subset of the training data. By segmenting the network's embedding layer into $D$ consecutive slices, we isolate $D$ distinct learners within the embedding space. Once these individual learner solutions converge, we aggregate them to reconstruct the entire embedding space. This merging process involves recombining the slices of the embedding layer that correspond to the $D$ learners. To ensure consistency in the embeddings produced by these diverse

[4]https://github.com/josharnoldjosh/ResNet-Extract-Image-Feature-Pytorch-Python

learners, we conduct fine-grained tuning across the entire dataset. However, when merging the embeddings, "shattered gradients problem" occurs, where the gradients resemble white noise, hindering training performance. To mitigate this challenge, *residual weights* are used [1] to provide the gradients with some spatial structure, which facilitates training, as illustrated in Figure 2.

### 4.3 Auto-Encoder Component

This part of the architecture enhances the model's ability to capture subtle nuances and opinions expressed in the input text. We leverage Context-Free-Grammar-Noun-Adjective-Pairs (Context Free ANP) to extract adjective-noun pairs from the utterances. This approach effectively allows our model to identify textual concepts. The ANP features, obtained through Context-Free ANP, are then fed into an auto-encoder to generate a latent representation. To infuse both textual and class semantic knowledge into the ANP representation, we employ an adversarial loss [38], as elaborated in the "Training and Inference" section. This adversarial loss is designed to disentangle syntax (captured by ANP) from semantics (captured by contextual character embeddings). This disentanglement enhances interpretability and offers better control over the learned representations.

### 4.4 Fusion

The Fusion component integrates information from both textual and visual modalities to provide a comprehensive understanding of the input data. To effectively fuse the knowledge from both modalities, we employ a shared fusion technique as shown in Figure 3. This technique uses cross-attention to capture the relationship between both modalities, preserving their unique characteristics.

4.4.1 *Shared-Features:* Assume that $F_a$, and $F_t$ correspond to the feature vectors of the image and text. As shown in Figure 3 (dotted box), $F_a$, and $F_t$ feature vectors are concatenated to provide the representation of the image and text features. Combined feature representation is given as:

$$Z : [F_a; F_t] \in \mathcal{R}^{D*L} \quad (1)$$

where D denoted by $D = x_a + x_t$ shows the concatenated features'(image, text) dimension. For the given multimodal review ($M_r$), the combined feature representations (Z) are now used to focus on the uni-modal feature representations $F_a$, and $F_t$. The combined features (Z) and the shared correlation matrix (M) between the image features are given by:

$$CoMat_a = tanh\frac{F_a^T W_{za} Z}{\sqrt{x}} \quad (2)$$

, where $W_{za} \in \mathcal{R}^{L*L}$ is the learnable weight matrix across image and shared image, and text features. Similarly correlation matrix for text features is defined as:

$$CoMat_t = tanh\frac{F_t^T W_{zt} Z}{\sqrt{x}} \quad (3)$$

The shared correlation matrices $CoMat_a$, and $CoMat_t$ for the image, and text modalities offer a semantic indicator of importance both within and between modalities. Within the same modality, there is a high correlation between the matching samples and the

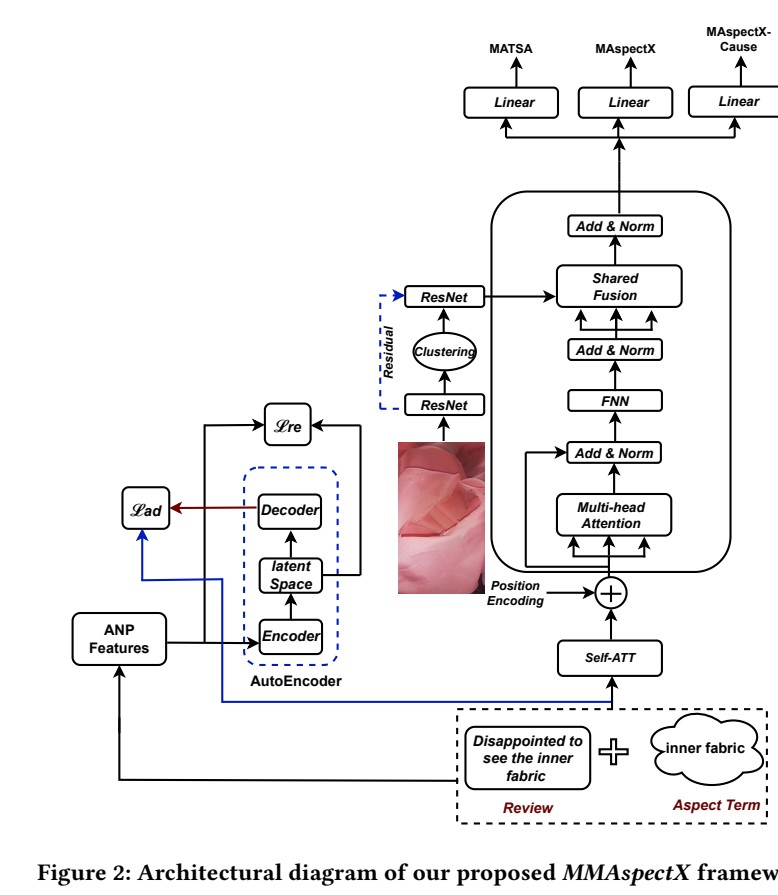

Figure 2: Architectural diagram of our proposed *MMAspectX* framework

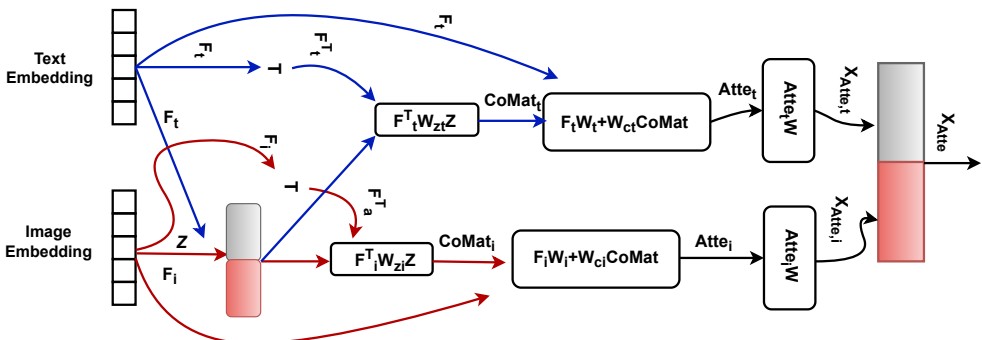

Figure 3: Shared Fusion Framework

other modalities, as indicated by a greater correlation coefficient of the shared correlation matrices $CoMat_t$, and $CoMat_a$. In order to improve the performance of the system, the suggested strategy effectively takes advantage of the complimentary nature of the image, and text modalities (i.e., inter-modal relationships) and intra-modal relationships. The attention weights of the image and text modalities are computed following the computation of the shared correlation matrices.

We use several learnable weight matrices corresponding to features of the separate modalities to compute attention weights for

the modalities because the dimensions of shared correlation matrices and the features of the associated modality vary. The learnable weight matrices $W_{ca}$ and $W_a$ are used to combine the shared correlation matrix $CoMat_a$ and the matching image features $F_a$, the following formula is used to calculate the attention weights for the image modality:

$$Atte_a = ReLu(W_a F_a + W_{ca} CoMat_a^T) \qquad (4)$$

, where $Atte_a$ represents the attention map of image modality. Similarly, for the textual modality, the equation is:

$$Atte_t = ReLu(W_t F_t + W_{ct} CoMat_t^T) \tag{5}$$

The attended characteristics of the image and text modalities are computed using the attention maps. These characteristics are attained by:

$$X_{Atte_a,a} = W Atte_a + F_a, \quad X_{Atte_t,t} = W Atte_t + F_t \tag{6}$$

Finally, the shared features of attended features of image and text are obtained by: $X_{Atte} = [X_{Atte_a,a}; X_{Atte_t,t}]$

### 4.5 Training and Inference

**Adversarial Loss:** Our goal is to reduce the gap between the discriminative capability of the text ($\theta(x)$ representing $SA(t)$) and the rich opinion structural information encapsulated in the feature $\phi(l_{po})$. This is accomplished by employing an adversarial constraint designed to deceive the discriminator network $\mathcal{D}$, thereby making the output features of $A(\theta(x))$ as similar to the ANP features as feasible. $\mathcal{L}_{ad} = \mathcal{E}_y(\log \mathcal{D}(h(y)) - \mathcal{E}_y(\log \mathcal{D}(\theta(y))$

**Joint Loss:** We train our model by incorporating a blend of the four diverse loss functions. $\mathcal{L}_{joint} = \mathcal{L}_{ad} + \mathcal{L}_{cause} + \mathcal{L}_{se} + \mathcal{L}_{ccp}$. In this context, $\mathcal{L}_{cause}$ denotes the loss associated with MAspectX-Cause, which is calculated using binary cross-entropy. Additionally, $\mathcal{L}_{se}$ and $\mathcal{L}_{ccp}$ represent the losses related to MATSA and MAspectX tasks, respectively, both of which are cross-entropy losses.

## 5 EXPERIMENTS AND RESULTS

In this section, we delve into the experiments we conducted, present the results we obtained, and provide a comprehensive analysis.

### 5.1 Experimental Setup

Our proposed model is developed using PyTorch[5], a Python-based deep learning framework. We employ the base version of BERT obtained from the huggingface transformers[6] package. All experiments are executed on an NVIDIA GeForce RTX 2080 Ti GPU. Our experiments run for 4 epochs, and we report the average scores after conducting 3 runs of the experiments to account for the inherent non-determinism associated with Tensorflow GPU operations. We utilize several evaluation metrics for the *MAspectX-Cause* task, including Full Match (FM), Partial Match (PM), Hamming Distance (HD), Jaccard Similarity (JS), and Ratcliffe-Obershelp Similarity (ROS). For *MATSA* and *MAspectX* tasks, we assess macro-F1 and Accuracy. We split the *MASpectX* dataset into an 80-20 ratio for training and testing. The best model is selected based on the performance on the validation set (10%)

**Baselines:** Our multimodal multi-task framework *MMAspectX* combines *MATSA*, *MAspectX*, and *MAspectX-Cause* into single system. Due to lack of suitable multimodal aspect based baselines with similar objectives (discussed in Section 2), chosen baselines includes the following systems: (i) Aspect category based multimodal complain detection system (ABCD) based on BiGRU with self attention layer [25], initially designed for binary classification task for aspect category based complaint detection (complaint vs

---

[5]https://pytorch.org/
[6]https://huggingface.co/docs/transformers/index

non-complaint); (ii) SpanBERT [9], MT-BERT (multi-task BERT) [18], and the Cascaded Multi-task System with External Knowledge Infusion (CMSEKI) using SenticNet [6]. We have extended these systems by including image and aspect term information in the input. Image and textual modalities are fused through concatenation in all these baselines. In line with the *MMAspectX* method, we have expanded the capabilities of these benchmark models to accommodate our multi-task scenario. More specifically, we have augmented them by adding two additional linear layers on top of the hidden-state outputs (for *MATSA* and *MAspectX*). For the output layer of the cause extraction task, we have incorporated a sigmoid activation function with a predetermined threshold value of 0.4. We have also compared our approach using few shot prompting using LLama textual model (Large Language Model Meta AI) [31] and LLava (multimodal) [14] models.

### 5.2 Experimental Results and Analysis

We investigate how various multimodal factors influence the tasks we are examining. The results of our proposed *MMAspectX* framework for *MAspectX* (F1), *MATSA* (F1), *MAspectX-Cause* (ROS) are depicted in Table 2. The bimodal configuration produces the most favorable results compared to unimodal networks (only text). Results illustrate the inadequacy of textual information alone in capturing all labels, necessitating the integration of image data to enhance feature representation. Our findings align with previous research [7]. Table 3 illustrates the effectiveness of multi-task learning. It shows that single tasks and dual tasks models perform inferior to models where all three tasks are solved jointly.

### 5.3 Comparison with Prior Works:

As depicted in Table 4, it is evident that CMSEKI stands out as the top-performing baseline. This result is in line with expectations, given that CMSEKI leverages common-sense knowledge from external sources to better comprehend input information. However, it is noteworthy that the proposed *MMAspectX* method consistently outperforms the CMSEKI model across all metrics, demonstrating a 3.92% and 2.14% improvement in F1 for the sentiment, and complain task, respectively, and a 3-point increase in the Ratcliffe-Obershelp Similarity (ROS) score. The relatively low performance observed in SpanBERT [9] highlights the challenges that even the powerful language models face in tackling critical tasks, such as cause extraction, particularly in situations involving complaints where training data may exhibit increased complexity. Furthermore, our observations indicate that user sentiment significantly enhances the performance of the *MMAspectX* method across the tasks under consideration. We perform the paired T-test (significance test), which validates that the performance gain over all the baselines is significant with 95% confidence (p-value<0.05). The *LLama* and *LLava* models also perform inferior to our proposed approach. It indicates that task-specific fine-tuning helps to learn domain information, which results in improvement.

### 5.4 Ablation Experiments

As shown in Table 4, we perform an ablation study on the *MAspectX* dataset to analyze the performance of the different components in our proposed framework. The values of the metrics for the

**Table 2: Results Across Various Modalities for MAspectX, MATSA, and MAspectX-Cause.**

| Modality | MAspectX | | MATSA | | MAspectX-Cause | | | | |
|---|---|---|---|---|---|---|---|---|---|
| | F1% | Acc. % | F1% | Acc. % | FM | PM | HD | JF | ROS |
| Text | 70.67 | 77.93 | 78.05 | 84.36 | 31.04 | 34.99 | 0.55 | 0.67 | 0.74 |
| Text+Image | **73.17** | **86.11** | **79.98** | **87.33** | **33.79** | **36.66** | **0.58** | **0.69** | **0.77** |

**Table 3: Experimental results of *MMAspectX* on various combination of tasks.**

| Tasks | MAspectX | | MATSA | | MAspectX-Cause | | | | |
|---|---|---|---|---|---|---|---|---|---|
| | F1% | Acc. % | F1% | Acc. % | FM | PM | HD | JF | ROS |
| MAspectX | 70.13 | 84.24 | - | - | - | - | - | - | - |
| MATSA | - | - | 72.10 | 84.01 | - | - | - | - | - |
| MAspectX-Cause | - | - | - | - | 30.68 | 33.37 | 0.54 | 0.64 | 0.74 |
| MAspectX+MATSA | 71.05 | 84.33 | - | - | - | - | - | - | - |
| MAspectX+MAspectX-Cause | - | - | 78.19 | 85.41 | 32.19 | 34.88 | 0.56 | 0.67 | 0.75 |
| MATSA+MAspectX-Cause | 71.31 | 84.22 | 78.68 | 85.24 | 32.43 | 34.79 | 0.56 | 0.68 | 0.75 |
| MAspectX+MATSA+MAspectX-Cause | **73.17** | **86.11** | **79.98** | **87.33** | **33.79** | **36.66** | **0.58** | **0.69** | **0.77** |

**Table 4: Comparison with prior works.**

| Models | MAspectX | | MATSA | | MAspectX-Cause | | | | |
|---|---|---|---|---|---|---|---|---|---|
| | F1 (%) | Acc. (%) | F1 (%) | Acc. (%) | FM | PM | HD | JF | ROS |
| *Baselines* | | | | | | | | | |
| **ABCD** [25] | 67.32 | 76.11 | 72.73 | 79.11 | 22.73 | 20.51 | 0.49 | 0.59 | 0.69 |
| **SpanBERTa** [9] | 69.81 | 80.22 | 75.76 | 84.22 | 26.58 | 23.43 | 0.53 | 0.62 | 0.72 |
| **MTL-BERT** [18] | 69.39 | 81.97 | 76.61 | 83.63 | 26.11 | 24.67 | 0.52 | 0.63 | 0.72 |
| **CMSEKI** [6] | 71.03 | 83.21 | 76.06 | 84.77 | 28.32 | 29.59 | 0.54 | 0.65 | 0.74 |
| **LLama** [31] | 45.73 | 74.77 | 54.57 | 77.24 | 22.89 | 24 | 0.36 | 48 | 0.50 |
| **LLava** [14] | 47.85 | 73.10 | 54.60 | 77.94 | 7.48 | 26 | 0.26 | 0.32 | 0.33 |
| **MMAspectX$_{Proposed}$** | **73.17** | **86.11** | **79.98** | **87.33** | **33.79** | **36.66** | **0.58** | **0.69** | **0.77** |
| *Ablation Study* | | | | | | | | | |
| **MMAspectX$_{-[SF]}$** | 71.25 | 81.35 | 77.50 | 85.84 | 31.22 | 34.74 | 0.56 | 0.67 | 0.74 |
| **MMAspectX$_{-[ANP]}$** | 71.83 | 84.59 | 77.84 | 86.19 | 31.73 | 34.97 | 0.56 | 0.67 | 0.75 |
| **MMAspectX$_{-[EC]}$** | 72.73 | 85.30 | 78.32 | 86.24 | 32.27 | 35.19 | 0.57 | 0.68 | 0.76 |
| **MMAspectX$_{-[SF+ANP]}$** | 70.77 | 83.09 | 75.36 | 84.01 | 30.54 | 33.55 | 0.55 | 0.66 | 0.73 |
| **MMAspectX$_{-[SF+ANP+EC]}$** | 69.44 | 81.69 | 74.02 | 83.66 | 28.93 | 32.79 | 0.54 | 0.65 | 0.73 |

*MAspectX*, *MATSA*, and *MAspectX-Cause* tasks are shown to drop when either the shared attention (*MMAspectX*$_{-SF}$), adjective-noun pair (*MMAspectX*$_{-ANP}$) or embedding clustering (*MMAspectX*$_{-EC}$) is omitted. The performance drop is more profound when either one, two, or all the components are removed. This affirms that the involvement of the shared fusion, adjective-noun pair, and embedding clustering of the utterances significantly contribute to the effectiveness of the three proposed tasks.

## 5.5 Human Evaluation

To manually evaluate the *MAspectX-Cause* task, 200 model-extracted spans are randomly selected and assessed based on established criteria. For each instance, we present the responses (produced by models and ground-truth by humans) to our annotators. The human raters hold post-graduate degrees in science and linguistics

and have prior annotation experience in text mining tasks. The assessment includes evaluating the spans for Fluency and Adequacy on a five-point scale, ranging from unacceptable (0) to excellent (5).

- **Fluency:** This metric evaluates the grammatical correctness of a sentence.
- **Adequacy:** Employed to ascertain if the generated response is meaningful and pertinent to the preceding conversation.

Additionally, the assessment considers the Informativeness of the extracted span. The Informativeness metric ranges from 0 (lack of information) to 5 (highly informative reasoning).

The results of the human evaluation, as presented in Table 6, align with the experimental findings discussed in Table 4. These findings affirm the superior performance of the *MMAspectX* model in comparison to the existing baselines. It is noteworthy that *MMAspectX* consistently outperforms the baselines across various manual evaluation metrics. The generated utterances are not only fluent but

**Table 5: Sample predictions from the various systems. Here, AT: aspect term.**

| Model | Text | AT | MAspectX | MATSA |
|---|---|---|---|---|
| 1. Human Annotator | Poor quality, inner fabric was torn when i get *inner fabric was torn when i get* | inner fabric | Complain | Neg |
| SpanBERT | Poor quality, inner fabric was *torn when i get* | | Concern | Neg |
| CMSEKI | *Poor quality, inner fabric was torn when i get* | | Complain | Neg |
| MMAsᴘᴇᴄᴛX (Proposed) | Poor quality, inner fabric was torn when i get *inner fabric was torn when i get* | | Complain | Neg |
| 2. Human Annotator | I bought this for my wife. *Fabric is of superb quality and stretchable also.* with out any doubt just go for it. | Fabric | Praise | Pos |
| SpanBERT | I bought this for my wife. Fabric is of superb quality and stretchable also. with out any doubt *just go for it.* | | other | Pos |
| CMSEKI | I bought this for my wife. *Fabric is of superb quality* and stretchable also. with out any doubt just go for it. | | Praise | Pos |
| MMAsᴘᴇᴄᴛX (Proposed) | I bought this for my wife. *Fabric is of superb quality and stretchable* also. with out any doubt just go for it. | | Praise | Pos |

**Table 6: Results of human evaluation on MAspectX-Cause task**

| Models | Fluency | Adequacy | Informativeness |
|---|---|---|---|
| SpaBERT | 2.21 | 2.29 | 2.31 |
| ABCD | 2.08 | 1.97 | 2.17 |
| CMESKI | 2.2 | 2.05 | 2.47 |
| MMAspectX | **3.06** | **3.21** | **3.23** |

also highly adequate, effectively capturing essential information from the user's perspective. The ratings collected from different annotators are averaged.

### 5.6 Qualitative Analysis

We conduct a thorough examination of predictions made by different systems, as illustrated in Table 5. In the top row, human annotators have identified tokens, referred to as 'causes,' which serve as representations of the causes for labels, such as *MATSA* and *MAspectX*. The subsequent four rows display tokens extracted by various models. It is evident that the proposed *MMAsᴘᴇᴄᴛX* model excels in accurately recognizing examples as instances of *MATSA* and *MAspectX*, providing high-quality causal spans. In contrast, the SpanBERT model, while correctly capturing a partial causal span, misclassifies the labels for both *MATSA* and *MAspectX*.

### 6 CONCLUSION

We introduced novel tasks, *multimodal aspect term based analysis beyond sentiment and complaint (MAspectX)* and *finding the cause of class predicted in the first task (MAspectX-Cause)*. We presented a multimodal dataset, *MAspectX*, that encompasses user complaints, praises, concerns, their associated sentiments, and the underlying causes behind all the classes at the aspect term level. This would immensely benefit organizations and enterprises in helping manage customer relationships by solving issues promptly and efficiently. We have developed a benchmark setup to create *MAspectX* corpus in a semi-supervised way using multi-step few-shot prompting followed by manual verifications of all the labels. Furthermore, we introduce a framework *MMAspectX* to solve *MATSA*, *MAspectX*, and *MAspectX-Cause* tasks jointly. Our results demonstrate that the proposed model *MMAspectX* can effectively share knowledge among all three tasks and outperforms various baselines.

### LIMITATIONS

Like most studies, this study has some limitations that could be addressed in future research. Our current study is focused on only a single cause of the *MAspectX* task. However, multiple causes can be present in the review that can provide more detailed information. These limitations could be addressed in the future by incorporating multiple causes in the dataset. The scope of our current work is confined to the English language, specifically within the clothing domain. In the future, our efforts will focus on enhancing our work in code-mixed languages and other domains. We would investigate ways of sharing knowledge between code-mixed and English languages to gauge the performance.

### ETHICS STATEMENT

This study has been evaluated and approved by our Institutional Review Board (IRB). The images and review text used to create the dataset for this study do not have any copyright clauses attached to them. Furthermore, the dataset will be shared (upon acceptance) via various channels for the main purpose of facilitating research and educational purposes.

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
