# OpenReview forum: "Aspect-Based Multimodal Mining: Unveiling Sentiments, Complaints, and Beyond in User-Generated Content"
_acmmm.org/ACMMM/2024/Conference — MM2024 Poster_

### Official Review · Reviewer_7S6v · 2024-05-24

**Rating:** 4
**Confidence:** 3

**Summary:**

The authors identified a gap in existing sentiment analysis and complaint recognition research, noting that the distinction between user concerns and complaints is not adequately refined. To address this, they propose a new task: aspect term-based multimodal analysis beyond sentiments and complaints. This task comprises two sub-tasks: multimodal aspect term-based analysis beyond sentiment and complaint (MAspectX) and identifying the cause of the class predicted in the first sub-task (MAspectX-Cause). For this new task, the authors use LLaVA to help construct a new multimodal dataset called MAspectX. Finally, they introduce the MMAspectX model and validate its effectiveness in addressing the new task using the new dataset.

**Strengths:**

a. The authors introduce a novel task along with the creation of a new dataset, offering a valuable resource for the multimodal sentiment analysis and complaint recognition community. This initiative not only enriches the available resources but also enhances the interpretability of tasks within this domain.

b. Key resources (sample dataset, source code) are available.

**Limitations:**

a. The authors would benefit from providing a more detailed definition and description of the inputs and outputs of the new task. This would greatly aid researchers in understanding and following the task effectively.

b. The paper requires thorough verification of writing details and diagrams：
(1) It appears that Z in Eq. (2) and Eq. (3) might serve as one of the subscripts of W.
(2) Throughout the paper, the subscript a denotes the image modal features, whereas in Figure 3, the subscript i is used for the image modal features. It is recommended to maintain a consistent subscript notation for clarity and coherence.
(3) According to Eq. (2) and Eq. (3), the steps to compute CoMat_a and CoMat_t should be symmetric. However, the computational flow depicted in Figure 3 does not appear to be symmetric.
(4) In Figure 2, there is a loss L_re, but in the final calculation of the joint loss, only L_se is included. What is the role of L_re?
(5) In Table 5, the text corresponding to SpanBERT and CMSEKI for the first case is incomplete.

c. In lines 394-395, how can the embedding dimension be divided into different groups with the image data? Isn't the embedding dimension a scalar?

d. What is the specific calculation process of  L_{re}? What is the motivation behind its design?

e. In Section 4.3, what is the theoretical support for the statement "separation of syntax and semantics enhances interpretability"?

f. In lines 624 to 626, the ratio of the training set to the test set is 8:2. Where is the validation set obtained from? In other words, what data did the author use for validation?

g. A key issue is that the MAspectX-Cause task is a generation task. So, how can it use a similar Linear layer to produce output as in the MATSA and MAspectX classification tasks? The author should provide an explanation of the decoding process for the MAspectX-Cause generation task, even if it's just a brief overview.

h. The experimental details need to be refined, such as the dropout rate, learning rate, early stopping, and embedding dimension.

**Suitability:**

3

---

### Official Review · Reviewer_zeqj · 2024-05-25

**Rating:** 4
**Confidence:** 3

**Summary:**

This paper proposes a novel task for multimodal aspect term-based analysis, which goes beyond traditional sentiment and complaint detection by introducing categories such as praise and concern. The authors develop a new benchmark dataset, MAspectX, and introduce a joint learning model (MMAspectX) for this task. The model aims to classify aspect terms into four classes (complaint, concern, praise, and others) and identify the causes behind these classifications. The experimental results show that the proposed model outperforms several baselines, demonstrating the effectiveness of incorporating multimodal information and joint learning.

**Strengths:**

(1) The paper introduces a novel task that extends traditional sentiment and complaint analysis by adding new categories like praise and concern.

(2) The creation of the MAspectX dataset, which includes multimodal (text and image) data annotated for aspect term-level complaints, praises, concerns, and their causes, is a significant contribution.

(3) The proposed MMAspectX model effectively integrates multiple tasks (sentiment analysis, aspect term classification, and cause extraction), demonstrating superior performance over existing baselines.

(4) The authors conduct extensive experiments and comparisons with various state-of-the-art models, providing strong evidence for the effectiveness of their approach.

(5) The paper includes a thorough analysis of the model's performance, including ablation studies and human evaluations, which enhance the credibility of the results.

**Limitations:**

(1) The dataset shows a strong dominance of positive/praise classes, which could bias the model's performance. How does the model perform on a more balanced dataset? Can you provide additional experiments to address this concern?

(2) While the authors employ manual corrections to ensure annotation quality, the reliance on large language models for initial weak labeling could introduce biases.

(3) You mentioned that the combination of text and images provides a better understanding of labels compared to a single modality, but I don't see a comparison with a single modality in Figure 1. Can you illustrate this difference?

(4) Can you modify Figure 2 to make the whole picture look more coordinated?

(5) Can you add some papers on the use of multimodal corpora (text + images) to analyze complaints at a fine-grained level, and need to analyze these methods?

(6) Why did you choose LLaVA as the aspect term extraction model? What are the different advantages of this model compared with other models?

(7) Notations and descriptions of the chart may need to be more detailed to ensure that the reader understands the information expressed in the chart even without reading the full text.

(8) Open-source code is recommended to ensure research transparency and reproducibility.

(9) While the paper compares the proposed model with several baselines, it does not include a comparison with some potentially relevant recent approaches in the field of multimodal sentiment and complaint analysis.

(10) The paper primarily focuses on standard metrics like F1-score and accuracy. Including more diverse evaluation metrics, such as precision, recall, and more detailed error analysis, could provide a more comprehensive understanding of the model's performance.

(11) While the paper mentions limitations, it lacks a detailed discussion on future work directions to address these limitations and further improve the model.

(12) The paper could provide more insights into how the relationships between text and image modalities are leveraged by the model and the impact of these relationships on performance.

**Suitability:**

2

---

### Official Review · Reviewer_ftmQ · 2024-05-25

**Rating:** 3
**Confidence:** 3

**Summary:**

The paper propose the task of multimodal aspect term-based analysis beyond sentiments and complaints to distinguish between concern and complaints, including classification of the given aspect term into one of four classes and identification of the cause of these classes. An effective technique is used for the joint learning of aspect term-based identification and cause extraction tasks(primary tasks), where sentiment analysis is used as a secondary task.

**Strengths:**

1.The paper proposes a new task named aspect term-based multimodal analysis beyond sentiments and complaints to bridge the gap between the differentiation between concerns and complaints and provide complaint analysis and corresponding cause extraction at the aspect term level.

2. It also builds a multimodal corpus (MAspectX) to solve both the tasks.

**Limitations:**

1.Figure2 and Figure3 are not very clear for understanding

2.The section of training and inference is not clear enough for understanding.

3.The section of experimental results and analysis is too brief.

**Suitability:**

2

---

### Official Review · Reviewer_5LEw · 2024-05-27

**Rating:** 4
**Confidence:** 3

**Summary:**

The paper addresses a significant gap in the existing literature by proposing a novel method for multimodal aspect-based analysis that distinguishes between complaints, concerns, and praise. This differentiation is particularly innovative as it goes beyond traditional sentiment analysis, providing more granular insights into user-generated content.

**Strengths:**

1. The introduction of the MAspectX and MAspectX-Cause tasks is commendable. The approach of using multimodal data (text and image) for classifying and extracting the cause behind each classification demonstrates a thorough and methodical advancement over unimodal analyses. This could potentially lead to more accurate and context-aware systems.
2. The creation of a new multimodal corpus specifically annotated for aspect term-based analysis is a significant contribution. This not only facilitates the specific tasks proposed but also sets a benchmark for future research in the area. Additionally, the use of multi-step few-shot prompting for annotation suggests an innovative approach to dataset creation, which could be of interest to other researchers dealing with similar challenges.

**Limitations:**

1.  While the paper introduces innovative tasks and a novel dataset, the description of the methodology lacks sufficient detail for full reproducibility. For instance, the specifics of the machine learning models used, feature extraction methods, and the exact nature of the multimodal integration are not thoroughly explained. More technical details would enhance the reader's understanding and the scientific robustness of the paper.
2. The paper would benefit from a more comprehensive validation section. While it claims that the proposed model outperforms state-of-the-art methods, there is limited discussion on the comparative analysis, such as the metrics used for benchmarking, statistical significance of the results, and the performance across different types of data (text-heavy vs. image-heavy).
3. The discussion section lacks an in-depth analysis of the limitations and potential biases inherent in the proposed methods and dataset.

**Suitability:**

2

---

### Official Review · Reviewer_ZVtd · 2024-05-27

**Rating:** 4
**Confidence:** 4

**Summary:**

This paper proposes a new task called "Multimodal Aspect Term-Based Analysis beyond Sentiment and Complaint (MAspectX)" and a related task "Multimodal Aspect Term-Based Cause Extraction (MAspectX-Cause)". The authors introduce a multimodal dataset (text and images) annotated for aspect term-based complaints, praises, concerns, their corresponding causes, and sentiments. They present a multi-task framework (MMAspectX) that jointly learns the proposed tasks, leveraging sentiment analysis as an auxiliary task. The paper claims to address the limitations of existing studies that lack fine-grained, aspect term-level analysis and the differentiation between complaints and concerns.

**Strengths:**

- The proposed tasks, MAspectX and MAspectX-Cause, address a gap in existing research by providing a more fine-grained analysis of aspect terms and distinguishing between complaints and concerns, which can be valuable for customer relationship management.
- The paper describes a semi-supervised approach using few-shot prompting and manual verification to create a multimodal dataset for the proposed tasks, addressing the lack of such resources.
- The proposed MMAspectX framework jointly learns the primary tasks (MAspectX and MAspectX-Cause) and the auxiliary sentiment analysis task, leveraging the complementary information from different modalities.
- The paper presents a thorough evaluation of the proposed framework, including comparisons with baseline methods, ablation studies, and human evaluations, demonstrating the effectiveness of the approach.

**Limitations:**

- The current work focuses on identifying a single cause for the MAspectX task, while multiple causes may exist in a review, limiting the level of detail and interpretability.
- The dataset and experiments are confined to the English language and the clothing domain, which limits the generalizability of the proposed approach to other domains and languages.
- While the paper introduces the concept of cause extraction, it does not provide in-depth analysis or visualization of the learned representations or the decision-making process, which could enhance interpretability and transparency.
- The paper compares the proposed approach primarily with baseline methods adapted from other tasks, as there are no directly comparable methods for the proposed tasks. A more comprehensive comparison with related tasks or approaches could provide additional insights.

**Suitability:**

3

---

### Meta-Review · Area_Chair_Ydep · 2024-07-02

**Recommendation:** Accept (Poster)
**Confidence:** 5

**Metareview:**

This paper presents a novel multimodal aspect-based analysis task that goes beyond traditional sentiment analysis by distinguishing between complaints, concerns, and praise. The authors introduce a new dataset and propose a joint learning model to address this task. While reviewers acknowledge the paper's contributions and innovative approach, they also point out several limitations, including the need for more methodological details, concerns about dataset bias, and questions about generalizability beyond the specific domain studied. The rebuttal appears to have addressed some concerns, leading one reviewer (ZVtd) to increase their score, while another decreased theirs. Despite these mixed reactions, the paper's novel task formulation and dataset contribution seem to outweigh its limitations. 4/5 reviewers have shown good interest and have leaned towards the acceptance of the paper. Given the potential impact on the field of multimodal sentiment analysis and my own insights into this work, I recommend accepting this paper, with the expectation that the authors will address the reviewers' concerns in the final version.